# Highly Sensitive Fingerprint Detection under UV Light on Non-Porous Surface Using Starch-Powder Based Luminol-Doped Carbon Dots (N-CDs) from Tender Coconut Water as a Green Carbon Source

**DOI:** 10.3390/nano12030400

**Published:** 2022-01-26

**Authors:** David Nugroho, Chayanee Keawprom, Saksit Chanthai, Won-Chun Oh, Rachadaporn Benchawattananon

**Affiliations:** 1Forensics Division, Department of Integrated Science, Faculty of Science, Khon Kaen University, Khon Kaen 40002, Thailand; davidnugraha95@gmail.com; 2Materials Chemistry Research Center, Department Chemistry and Center of Excellence for Innovation Chemistry, Faculty of Science, Khon Kaen University, Khon Kaen 40002, Thailand; chayanee.ka@kkumail.com; 3Department of Advanced Materials Science and Engineering, Hanseo University, Seosan-si 356-706, Korea; wc_oh@hanseo.ac.kr

**Keywords:** carbon dots, latent fingerprints, luminol, chemiluminescence, forensic science, nanomaterial

## Abstract

This study aims to synthesize carbon dots from a natural resource and will be used to detect a latent fingerprint on a non-porous surface. The carbon dots (CDs) were prepared by adding luminol to coconut water and ethanol via a hydrothermal method. Luminol enhances the chemiluminescence of the CDs, which show more distinct blue light under a UV lamp compared with bare CDs. To detect the latent fingerprint, luminol carbon dots (N-CDs) were combined with commercial starch and stirred at room temperature for 24 h. Their characteristics and optical properties were measured using EDX-SEM, HR-TEM, FTIR, XPS, UV–visible absorption, and fluorescence. In this research, it was found that the N-CDs had a d-spacing of 0.5 nm and a size of 12.9 nm. The N-CDs had a fluorescence intensity 551% higher than the standard normally used. N-CDs can be used to detect latent fingerprints on a non-porous surface and are easy to detect under a UV lamp at 395 nm. Therefore, luminol has a high potential to increase sensitive and stable traces of chemiluminescence from the green CDs for forensic latent fingerprint detection.

## 1. Introduction

Fingerprints are always used as the gold standard for identifying someone, especially in the forensic field [1,2] and are currently widely used as biometrics and scientific evidence in courts of law and the police field. Papillary protrusions on the tips of the fingers, which contain rows of pores connected to the sweat glands, make each person’s fingerprints unique. The only exception would be a person who had been in an accident so serious that the papillary tissue on the fingers, palms, soles of the feet, and feet were damaged [3]. Because fingerprints are unique to each person, cannot be changed, are easy to verify, and leave marks on every object a person touches [4], they are routinely used as evidence in court and by the police. In a criminal case, investigators will usually test for fingerprints on objects found at the crime site because they are very important evidence during an investigation [5]. There are generally three types of fingerprints found at the crime site: plastic fingerprints, patent fingerprints, and latent fingerprints. To detect latent fingerprints, investigators need some physical or chemical processes (water, amino acids, oils, and some other substances) to enhance the fingerprint residue because latent fingerprints cannot be seen by the naked eye [6]. 

Carbon dots (CDs) are a new class of carbon nanomaterials with sizes below 10 nm and have a three-dimensional structure in the nano regime [7,8,9,10,11,12,13]. The general structure of CDs is sp^3^/sp^2^, which means CDs can use a fluorescence property that can be applied in various fields such as bioimaging, biosensors, pH sensing, and biomolecule or drug delivery. Due to the flexibility of the PL emission wavelength of CDs, they can be used as an alternative material in nanoclusters and traditional dye molecules in the optoelectronics field. The PL property of CDs depends on the source of carbon, synthesis method, and parameter-setting preparation. The CDs have wide potential in optronics, catalytics, and sensors because of their outstanding electronic properties due to carbon-based quantum dots as donors and acceptors as electrons, which causes chemiluminescence and luminescence [14,15,16,17,18,19,20,21,22,23,24,25,26,27,28,29,30]. Heteroatoms, such as nitrogen, can provide additional electrons and can increase emission brightness. Previous research has shown some sources for synthesis of N atoms such as freshly prepared pear juice and ethanediamine [31], sucrose and urea [32,33], chitosan and ammonia [34], and polyvinylpyrrolidone (PVP) [35].

The synthesis of CDs can be conducted via the green synthesis route by employing natural materials, such as watermelon peel, orange, red beetroot, etc., as starting carbon sources. The synthesis of CDs is roughly classified in two ways: “top-down” and “bottom-up”. Top-down synthesis breaks the large carbon materials into small segments by using processes such as laser ablation and chemical oxidation. In bottom-up synthesis, the carbon precursors use methods such as hydrothermal, pyrolysis thermal, and microwave [36,37,38,39,40,41,42,43,44,45,46,47].

In our report, we used the green synthesis of carbon dots from coconut water because coconut water is a natural product, rich in carbon and nitrogen, and a source of approximately 5.23 g/100 g of sugars, as shown in Table 1 [48]. In this research, coconut water was heated with the hydrothermal method. After the carbon quantum dot liquid was ready, in the next step we mixed it with commercial starch at room temperature for 24 h using a stirrer and dried it for 24 h in oven chamber at 60 °C.

## 2. Results and Discussion

### 2.1. Spectroscopic Characteristics of the N-CDs by XPS and FTIR

X-ray photoelectron spectroscopy (XPS) of N-CDs was analyzed to confirm the surface functionality of the N-CDs. The survey spectrum as shown in Figure 1a reveals the existence of carbon (C 1s, 284.7 eV), nitrogen (N 1s, 399.0 eV), and oxygen (O 1s, 531.7 eV) with the percentage of atomic C1s, N1s, and O1s at 64.2%, 8.5%, and 27.4%, respectively. Deconvolution of C1s showing energy of 284.7 eV is the binding energy of the C-C bond, of the C-O bond with energy of 285.4 eV, and of the C=O with energy of 286.4 eV (Figure 1d). Nitrogen from luminol doping was detected by XPS analysis as a C-N bond formation with binding energy of 399.0 eV (Figure 1b). The O1s peak is shown in Figure 1c. Upon deconvolution, peaks at 532.0 eV are attributed to O=C and to O-C in the energy peak 530.8 eV [49,50].

Functional groups of N-CDs were investigated by FTIR analysis as shown in Figure 2. Figure 2a shows that the O-H stretching band appears at about 3300 cm^−1^, the alkane bond of the C-H bond at 2920 cm^−1^, strong appearance of the C=O bond at 1700 cm^−1^, and also of the C-O stretching bond at 1194 cm^−1^ for coconut water carbon dots [50]. C-O also was found at 1022 cm^−1^ [51], and 774 cm^−1^ was of CH_2_ stretching bonds. Figure 2b shows the FTIR from the N-CDs sample and found the N-H stretching bond at about 3398 cm^−1^ [52], the C-H stretching alkene bond at 2916 cm^−1^, the C=O stretching of ester groups between 1597 and 1656 cm^−1^, the C=C bond [53,54] at 1494 cm^−1^, the C-N stretching bond at 1244 cm^−1^, the C-O stretching bond at 1048 cm^−1^, and the C-H bond at 781 cm^−1^ [55].

### 2.2. Morphology Image Analysis of N-CDs by FIB-SEM and HR-TEM

Figure 3a is an image analysis by a high-resolution transmission electron microscope (HR-TEM), which was used to confirm the FIB-SEM image obtained above. The usual characteristic diameter of the N-CDs was about 12.9 nm with the d-spacing of 0.5 nm. Under the focused ion beam scanning electron microscopy (FIB-SEM) the surface of N-CDs appears to be completely coated with luminol, as compared in Figure 3c,d. Synthesis of the carbon dots with luminol (N-CDs) has an uneven textured surface structure, and N-CDs mixed with tapioca flour also give a coarser texture as shown in Figure 3b. It also affected the basic properties of the carbon dots. The carbon dots from coconut water synthesized without using luminol are easier to melt at room temperature, as shown in Figure 3d.

### 2.3. Elemental Analysis of the N-CDs by EDX

Figure 4 compares characteristics of CDs and N-CDs under EDX. Figure 4a shows the elemental composition of CDs as C 62.6% and O 35.6%, compared with the elemental composition of N-CDs as C 54.1% and O 44.2% (Figure 4b).

### 2.4. Optical Properties of CDs and N-CDs

Optical properties of CDs were analyzed by UV–visible absorption, fluorescence, and common UV lamp. The synthesis of carbon dots from coconut water was carried out following the previous ratio, 1:1 *v*/*v* (ethanol:coconut water). This was slightly modified by various durations (1, 2, 3, and 4 h) and temperatures (140, 160, 180, and 200 °C). Then the CDs were doped with luminol to become N-CDs by adding 1 g luminol solid under the suitable preparation of the CDs. The optical properties of CDs that were synthesized at different durations and temperatures were monitored. In this study it was shown that pure coconut water did not have luminescence properties under a UV lamp. However, after being heated using the hydrothermal methods, CDs showed luminescence behavior as a function of the reaction temperature and the duration differences. 

The synthesis procedure of CDs with varying durations was conducted under a UV lamp, and its appearance was observed as shown in Figure 5. Briefly, under UV light the energy of the maximum wavelength at 380 nm excites the ground state electron of the CDs, and then the excited electron undergoes an energy transfer, which is visible as a fluorescence spectrum. The CDs product obtained from 1 h showed no blue light under UV light; at 2–3 h it was a yellow light emission, and at 4 h it had more blue light and a yellowish-brown color in visible light compared with the others. The UV–visible absorption spectrum and photoluminescence intensity of the CDs was recorded. The spectral data on the effect of various synthetic durations of CDs are shown in Figure 5a. It was found that a 4 h hydrothermal incubation has a high absorbance value in the UV–visible range and high fluorescence intensity as well (Figure 5b).

The comparison of fluorescence spectra with various temperatures is shown in Figure 6. It was evident that the synthesized CDs had a high peak emission wavelength at 465 nm, particularly at 200 °C, and had an intensity higher than others. However, they only had a small gap in intensity compared with 180 °C, while at 180 °C they had more blue light under UV light, and a yellowish-brown color in visible light (Figure 6a). This research uses the optimum data (4 h and 180 °C) from this synthetic route. 

Figure 7 shows that the fluorescence intensity of luminol and coconut water (N-CDs) was 551% higher than luminol carbon dots (NDs). Compared to raw luminol, CDs (synthesis from coconut water) or NDs (synthesis from luminol), N-CDs (synthesis from coconut water and luminol) had a very high fluorescence intensity. Figure 8 shows that under UV light, the N-CDs had a bright blue color compared to others. The effect of a pH buffer solution on the fluorescence properties of the N-CDs was also investigated in detail. The N-CDs were also compared with previous literature reports (Table 2).

One hundred µL N-CDs was added into 1 mL of pH buffer solution ranging from 2 to 12, then adjusted to 10 mL using deionized water as shown in Figure 9. It was found that an acidic pH of 2 to 6 can increase the fluorescence intensity of the N-CDs, with the most optimum condition being pH 4. In addition, it was evident that at pH 7 to pH 12, which is a basic solution, the fluorescence intensity of the N-CDs can be reduced.

### 2.5. Application of the N-CDs for Latent Fingerprint Detection

Tapioca flour combined with N-CDs has fluorescence under UV light. Tapioca flour can bind with the fat in fingerprint residue and act as substrate to detect a latent fingerprint on non-porous material. To test the durability of the CDs powder, we utilized it to detect a latent fingerprint on non-porous surface material. In the present study, the results of various concentrations of the CDs on commercial tapioca flour powder are shown in Figure 10. CDs of 20 mL were brighter and had a more detailed minutiae fingerprint result compared with the others under UV light. A solution of 0.1 g of each CDs powder was diluted with 100 mL DI water, to which 1 mL in 9 mL of DI water was added to check the fluorescence intensity. Figure 10e shows that 20 mL of CDs with 5 g starch gives higher fluorescence intensity compared to others.

CDs powder was compared with N-CDs powder in the same condition with most suitable concentrations of the CDs powder (20 mL CDs with 5 g starch). Figure 11a shows that the fluorescence intensity of the N-CDs was higher compared with those of the others. Figure 11b shows that under UV light the latent fingerprint that can be detected using N-CDs was brighter compared with bare CDs.

To detect the latent fingerprints on the non-porous surface in various surface materials, after applying the N-CDs powder on the surface, the surface was checked under a UV lamp to compare the contrast in the latent fingerprints. The results in Table 3 show that N-CDs powder can be used to detect latent fingerprints on many types of surfaces, such as glass, coin, and plastic, under UV light.

To test the resistance and durability of the carbon dots powder, the study compared the N-CDs powder that was synthesized on the first day with the N-CDs powder that was kept for 30 days and stored at room temperature. Figure 12a shows that the N-CDs powder could still be used to detect latent fingerprints even though the quality of the fluorescence light produced was not as bright as on the first day of the synthesis (Figure 12b).

We also tested how storage at different temperatures and duration affected the ability to find latent fingerprints. A volunteer left a fingerprint on a surface. We kept it at −4 °C and at 60 °C in the Forensic Science department, Khon Kaen University. We kept the sample at −4 °C before checking it using CDs powder. The material surface was dry and heated in a 60 °C oven for 30 min. Table 4 shows that N-CDs can be used to detect latent fingerprints on many material surfaces, with DM meaning that minutiae in the fingerprint are detected, D meaning that the fingerprint is detected but that the minutiae are not clear, and ND meaning that the fingerprint is not detected.

## 3. Materials and Methods

### 3.1. Chemicals and Reagents

Commercial tapioca flour (Five Stars Fish brand) was purchased from a local supermarket (Khon Kaen, Thailand). Coconut water was purchased from the retail food market, Khon Kaen University (Khon Kaen, Thailand). Ethyl alcohol anhydrous 99.9% was obtained from Daejung Chemicals & Metals Co., Ltd. (Siheung, Korea).

### 3.2. Instruments

The synthesis of CDs was carried out in a heating oven (BINDER ED 115UL-120V). Centrifuge NF 800 and magnetic stirrer (Daihan Scientific MSH-20D) were used for the mixture with commercial tapioca powder. The CDs product was checked by an optical property (under commercial UV Lamp at 365 nm), fluorescence, and UV–visible spectrophotometers, and characteristics under X-ray photoelectron spectra (KRATOS AXIS SUPRA), FTIR (Fourier Transform Infrared Spectrometer (FT-IR) Bruker TENSOR 27), EDX (Oxford Instruments X-Maxn), FIB-SEM (Helios NanoLab G3 CX) and HR-TEM (Spectrometer: Transmission Electron Microscope; Electron gun: Schottky field emission type electron gun).

### 3.3. Synthesis of the Luminol Doped Carbon Dots (N-CDs)

Twenty mL of coconut water was filtered using a nylon filter membrane 0.22 µm and mixed with 20 mL ethyl alcohol anhydrous (1:1). The synthesis process was performed by a hydrothermal method, by heating in an oven with different temperatures (200, 180, 160, and 140 °C) and different reaction times (1, 2, 3 and 4 h). After synthesis, the product was centrifuged at 10,000× g rpm for 10 min. After the optimum conditions of CDs were obtained, 1 g luminol powder was added through the synthetic step (20 mL tender coconut water: 20 mL ethyl alcohol: 1 g luminol powder) by the hydrothermal method using a 180 °C oven for 4 h, then cooling down at room temperature for 24 h, and centrifuging at 10,000× g rpm for 10 min.

### 3.4. Fluorescent N-CDs Powder Preparation

Five grams of the commercial tapioca flour powder was mixed with various concentrations of the CDs (5, 10, 15, and 20 mL) for 24 h. The mix was magnetically stirred at room temperature at 400 rpm. Then it was kept dry in a 60 °C oven to become powder. The powder was then stored for further use at room temperature.

### 3.5. Method Detection of a Latent Fingerprint

The latent fingerprint was detected by the common brushing method. A volunteer rubbed his forehead with a finger and printed his fingerprint on the different material surfaces for 3 s. The N-CDs powder was brushed on the fingerprint. Finally, the latent fingerprint was detected under a UV lamp.

## 4. Conclusions

The CDs synthesized from coconut water and starch produce blue color under a UV lamp. To create the optimum conditions in this research we added luminol to improve the color intensity. N-CD characteristics and optical properties were observed using EDX-SEM, HR-TEM, FTIR, XPS, UV–visible absorption, and fluorescence, with the diameter of the N-CDs around 12.9 nm and a d-spacing of 0.5 nm. The solution application was carried out with experiments on various non-porous materials (glass, plastic, etc.). The experiments also measured the effect of temperature and storage time of the material and the effect of storage time of carbon dots powder for 30 days.

## Figures and Tables

**Figure 1 nanomaterials-12-00400-f001:**
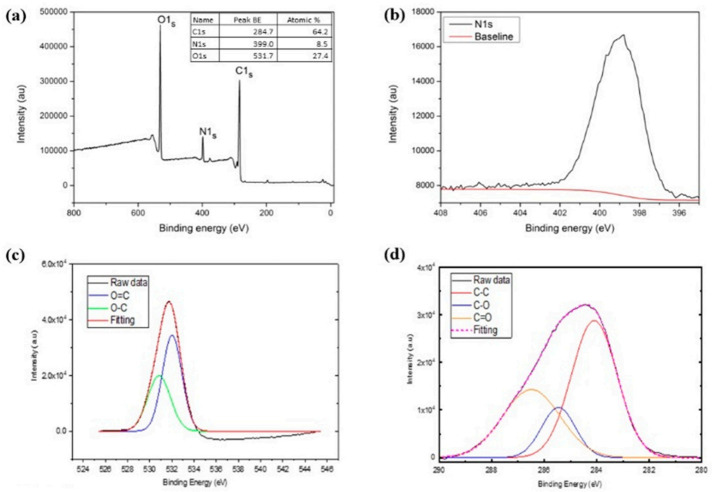
XPS analysis of N-CDs: (**a**) survey spectrum, (**b**) N1s, (**c**) O1s, and (**d**) C1s.

**Figure 2 nanomaterials-12-00400-f002:**
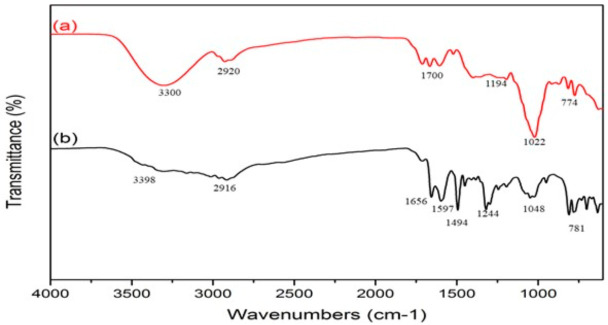
Fourier transform infrared spectroscopy (FTIR) of (**a**) CDs and (**b**) N-CDs.

**Figure 3 nanomaterials-12-00400-f003:**
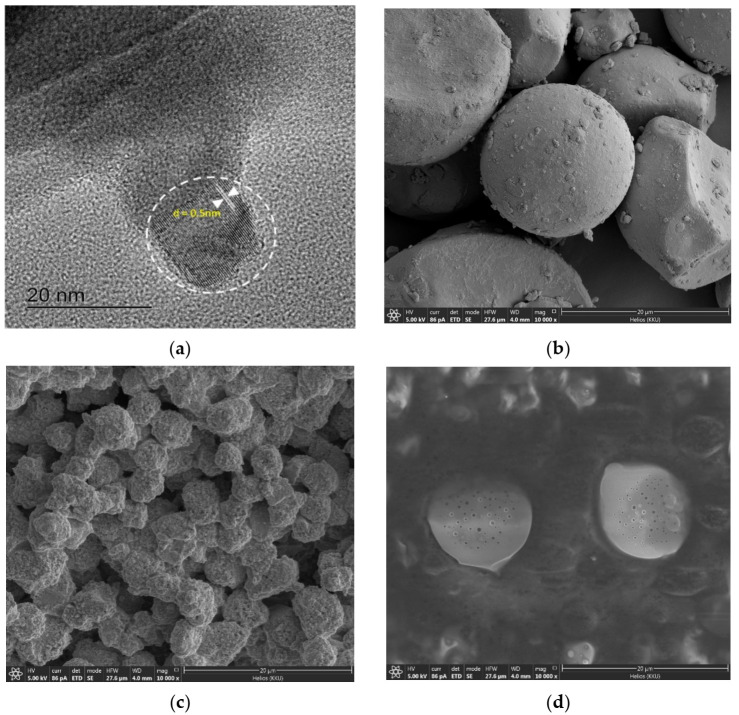
(**a**) High-resolution-transmission electron microscope (HR-TEM) of N-CDs. Focused ion beam-scanning electron microscopy (FIB-SEM) of (**b**) N-CDs/starch, (**c**) N-CDs, and (**d**) CDs.

**Figure 4 nanomaterials-12-00400-f004:**
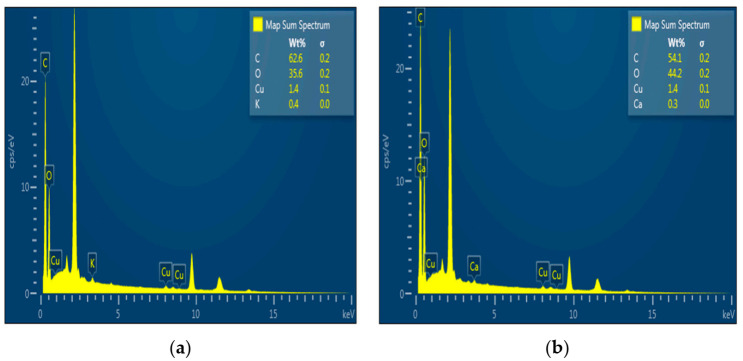
EDX analysis of (**a**) CDs and (**b**) N-CDs.

**Figure 5 nanomaterials-12-00400-f005:**
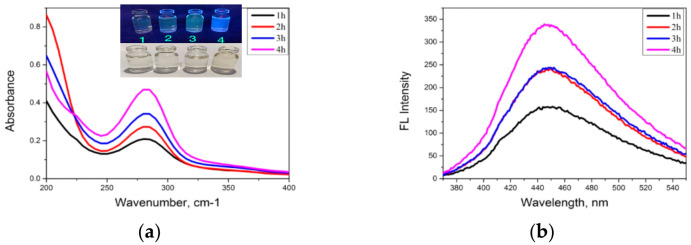
(**a**) UV–visible absorption spectra of CDs and (**b**) fluorescence spectra of CDs with various durations.

**Figure 6 nanomaterials-12-00400-f006:**
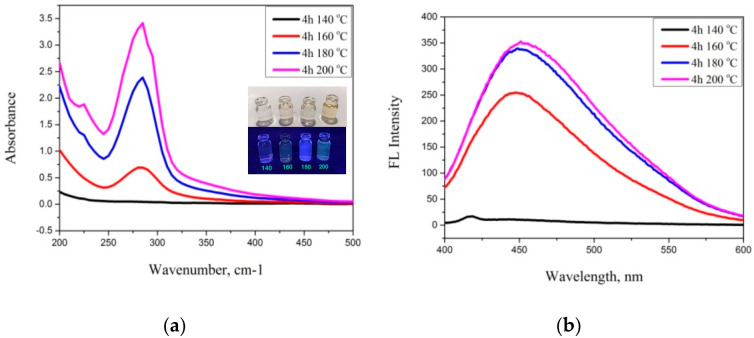
(**a**) UV–visible absorption spectra of various temperatures of CDs and (**b**) fluorescence spectra of CDs with various temperatures.

**Figure 7 nanomaterials-12-00400-f007:**
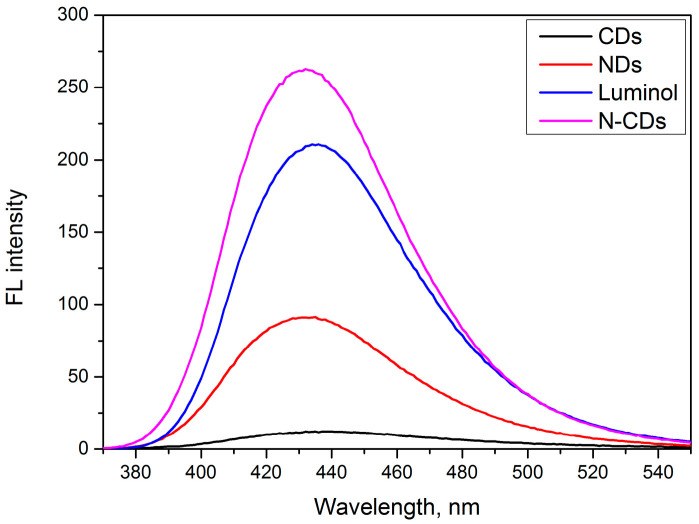
Fluorescence spectra comparison of CDs, N-CDs, raw luminol, and NDs.

**Figure 8 nanomaterials-12-00400-f008:**
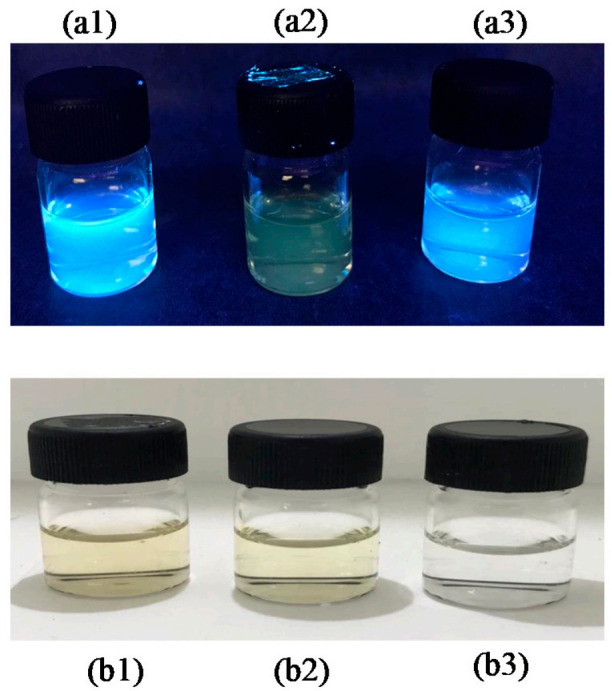
UV light analysis: (**a1**) UV chamber of N-CDs, (**a2**) UV chamber of CDs, and (**a3**) UV chamber of NDs. Visible light: (**b1**) visible light of N-CDs, (**b2**) visible light of CDs, and (**b3**) visible light of NDs.

**Figure 9 nanomaterials-12-00400-f009:**
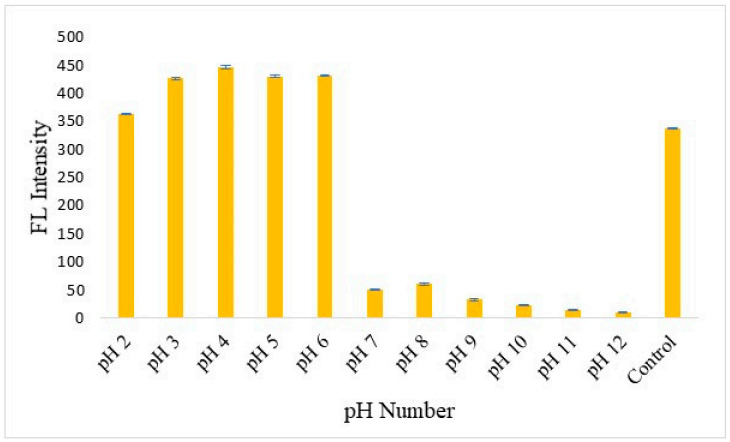
Fluorescence spectra comparison of pH buffer solution of 2–12 in N-CDs. Error bars represent standard deviations of three independent measurements.

**Figure 10 nanomaterials-12-00400-f010:**
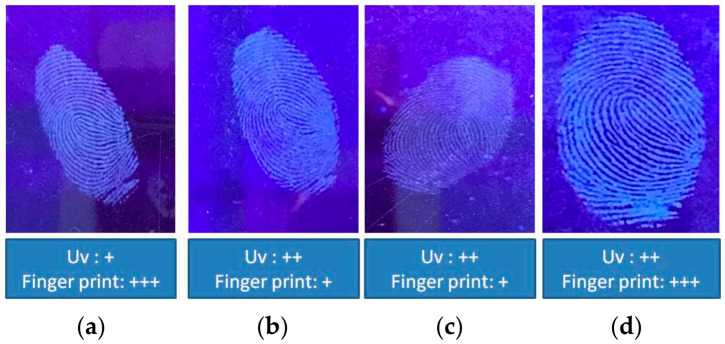
UV lamp analysis with various concentrations of CDs on starch: (**a**) 5 mL/5 g starch; (**b**) 10 mL/5 g starch; (**c**) 15 mL/5 g starch; (**d**) 20 mL/5 g starch; (**e**) fluorescence spectra for comparative concentrations of CDs on starch.

**Figure 11 nanomaterials-12-00400-f011:**
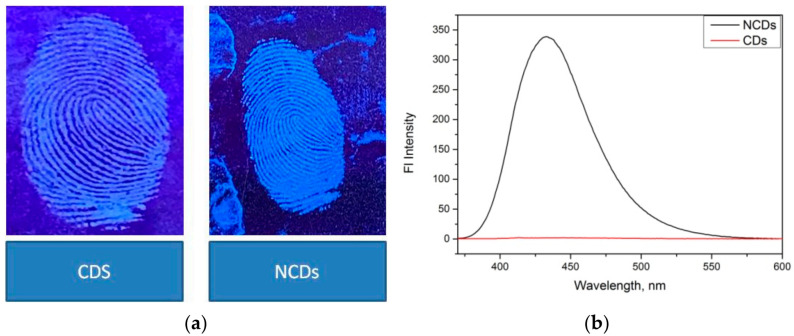
(**a**) UV lamp analysis of CDs vs. N-CDs and (**b**) fluorescence spectra for comparison of N-CDs vs. CDs.

**Figure 12 nanomaterials-12-00400-f012:**
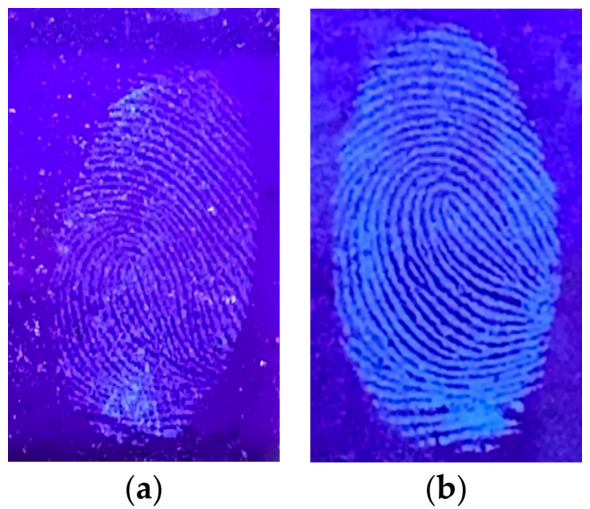
Effect of storage time on the ability to detect a latent fingerprint: (**a**) 30th day after synthesis of N-CDs; (**b**) 1st day after synthesis of N-CDs powder.

**Table 1 nanomaterials-12-00400-t001:** Some chemicals of coconut water.

Coconut Type	Young	Mature
Average Weight (g)	565	393
Age of Coconut	6 months	12 months
sugar	(g/100 g)	
Total	5.23	3.42
Sucrose	0.06	0.51
Glucose	2.61	1.48
Fructose	2.55	1.43
Inorganic Ions	(g/100 g)	
Calcium, Ca	27.35	31.64
Potassium, K	203.7	257.52
Phosphorus, P	4.66	12.77
Sodium, Na	1.75	16.1

**Table 2 nanomaterials-12-00400-t002:** Fluorescence observed from different carbon dots prepared from various precursors.

Precursors	Reaction Condition	Fluorescence	Size	Application	Reference
Sodium alginate	1100 W for 8 min	Blue	5.6 nm	Toughening agent	[33]
Watermelon peel	220 °C for 2 h	Blue	2 nm	Optical imaging	[39]
Table sugar	120 °C for 3 min	Blue	3–4 nm	Green reducing agent	[40]
Red beetroot	180 °C for 10 h	-	5–7 nm	Pd^2^^+^ detection	[41]
Green tea leaf	350 °C for 2 h	Blue	2 nm	Optical imaging	[42]
Orange juice	120 °C for 150 min	yellow	1.5–4.5 nm	Biolabeling and optoelectronics	[43]
Aloe	180 °C for 11 h	Blue	5 nm	Detection of tetrazine in food samples	[44]
Orange pericarp	100 °C for 5 h	Blue	2.9 nm	Nano biotechnology	[45]
Cabbage	140 °C for 5 h	Blue	2–6 nm	Bioimaging	[46]
Coconut water	180 °C for 4 h	Blue	12.9 nm	Detection fingerprint	Our work

**Table 3 nanomaterials-12-00400-t003:** Application of the obtained N-CDs powder for latent fingerprint detection on some non-porous material surfaces.

Material Type	Visible Lamp	UV-Light
Glass	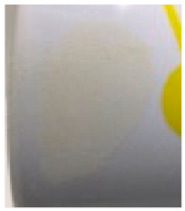	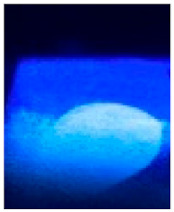
Stapler	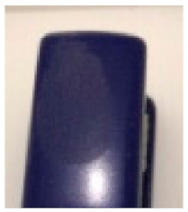	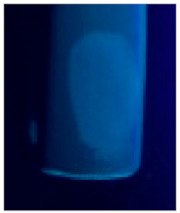
Coin	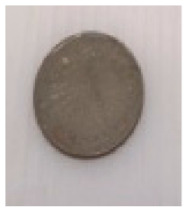	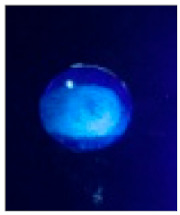
Coaster	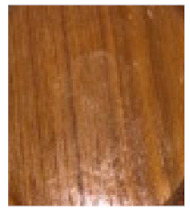	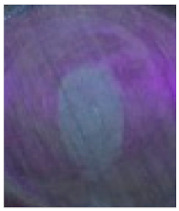

**Table 4 nanomaterials-12-00400-t004:** Effect of storage time and temperature on the latent fingerprint application on glass, iron, and plastic surfaces.

Material	Day	60 °C	−4 °C	Room Temp
Glass	7	DM	D	DM
	14	D	D	DM
	21	D	D	DM
	28	D	D	DM
Iron	7	DM	DM	DM
	14	DM	DM	DM
	21	DM	D	DM
	28	D	D	DM
Plastic	7	D	D	DM
	14	D	D	DM
	21	D	D	D
	28	D	D	D
D	Detected			
DM	Detected Minutiae			
ND	Not Detected			

## Data Availability

Data can be available upon request from the authors.

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
