# Peer review of "Highly Sensitive Fingerprint Detection under UV Light on Non-Porous Surface Using Starch-Powder Based Luminol-Doped Carbon Dots (N-CDs) from Tender Coconut Water as a Green Carbon Source"

_nanomaterials, 2022, doi:10.3390/nano12030400_

Round 1

Reviewer 1 Report

The manuscript entitled, ‘Highly sensitive fingerprint detection under UV light on non-porous surface using starch powder based luminol doped carbon dots (N-CDs) from tender coconut water as a green carbon source’ discussed the activity of NCDs for finger print detection. The work is quite interesting and summarized the leading areas. I am mentioning some loopholes of this work which should be accounted prior to publication;

  1. First of all, the aim of the work should be rewritten.
  2. Why coconut water is taken in not clear. It will be better to incorporate some special features or minerals present in coconut water.
  3. The crude obtained after hydrothermal looks black. Did the author dialysis it or directly used it?
  4. Some articles related to carbon dots will improve its results and discussion and introduction part. I am mentioning some of them: 10.1021/acsami.0c14527; 10.1002/slct.202001613.
  5. Figure 11 should be modified with error bar.

Author Response

Response to reviewer’s comments

Journal: MDPI-Nanomaterial

Manuscript ID: nanomaterials-1565490

Title: Highly sensitive fingerprint detection under UV light on non-porous surface using starch powder based luminol doped carbon dots (N-CDs) from tender coconut water as a green carbon source

Dear Editor of Nanomaterial - MDPI

Thank you for your useful comments and suggestions of our manuscript. We have modified the manuscript accordingly, and the detailed corrections are listed:

Response to Comments from Reviewer #1

The authors thank the reviewer for the comments. The revision is highlighted in yellow.

Question or Comment and Answer and Suggestion:

  1. First of all, the aim of the work should be rewritten

A1: The aims have been rewritten become,

“This study aims to synthesize the carbon dots from natural resource and will be applied for detection of latent fingerprint on non-porous surface material, the carbon dots (CDs) prepared by using coconut water, ethanol, and adding luminol process via hydrothermal method”

  1. Why coconut water is taken in not clear. It will be better to incorporate some special features or minerals present in coconut water

A2: The reason why using coconut water including the suggestion about minerals had been adding in the revised MS accordingly.

  1. The crude obtained after hydrothermal looks black. Did the author dialysis it or directly used it?

A3: The author directly uses the carbon dots after fresh synthetic products.

  1. Some articles related to carbon dots will improve its results and discussion and introduction part. I am mentioning some of them: 10.1021/acsami.0c14527; 10.1002/slct.202001613

A4: All of these suggested references had been reviewed and the interpreting in the revised MS accordingly.

  1. Figure 11 should be modified with error bar.

A5: Figure 11 had been modified with the error bar and adding in the revised MS accordingly.

The revised manuscript has been once resubmitted to your journal. We look forward again to your positive response. Should you have any doubts, please do not hesitate in contacting us.

Looking forward to hearing from you.

Yours sincerely,
S. Chanthai

(Prof. Dr. Saksit Chanthai)

Reviewer 2 Report

In this paper, carbon dots (CDs) were synthesized from coconut water and doped by luminol. The obtained CDs were used to detect the latent fingerprint by combining with commercial starch. This work is interesting and puts forward a new venue for fingerprint detection presenting a potential application in forensic field. Before considering for publication, the authors need to make a major revision on current manuscript.

  1. As the authors mentioned that CDs were synthesized by employing various natural materials, such as fennel seeds, ginger and so on. It is suggested that compare the properties of the CDs synthesized in this work with the CDs synthesized by other natural materials.
  2. The legends in Fig. 3b, 3c and 3d were invisible to be seen. The difference between N-CDs/starch and N-CDs was difficult to be recognized.
  3. The photoluminescent mechanism of the CDs obtained in this work was not clearly explained.
  4. Grammar mistakes and typos were often found in the manuscript, please double check the whole text and polish the language so that the manuscript is more readable.

Author Response

Response to reviewer’s comments

Journal: MDPI-Nanomaterial

Manuscript ID: nanomaterials-1565490

Title: Highly sensitive fingerprint detection under UV light on non-porous surface using starch powder based luminol doped carbon dots (N-CDs) from tender coconut water as a green carbon source

Dear Editor of Nanomaterial - MDPI

Thank you for your useful comments and suggestions of our manuscript. We have modified the manuscript accordingly, and the detailed corrections are listed:

Response to Comments from Reviewer #2

The authors thank the reviewer for the comments. The revision is highlighted in Blue.

Question or Comment and Answer and Suggestion:

  1. As the authors mentioned that CDs were synthesized by employing various natural materials, such as fennel seeds, ginger and so on. It is suggested that compare the properties of the CDs synthesized in this work with the CDs synthesized by other natural materials.

A1: The comparison between previous research with our research has been adding in our revised MS as shown in Table 2.

  1. The legends in Fig. 3b, 3c and 3d were invisible to be seen. The difference between N-CDs/starch and N-CDs was difficult to be recognized.

A2: The author has been trying to change the picture of the results to be more easily to recognize and improve the legends and the interpreting in the revised MS.

  1. The photoluminescent mechanism of the CDs obtained in this work was not clearly explained.

A3: The photoluminescent mechanism of the CDs was briefly added in the revised MS (line 233 – 235)

  1. Grammar mistakes and types were often found in the manuscript, please double check the whole text, and polish the language so that the manuscript is more readable.

A4: After the manuscript was revised, it has been grammatically read and discussed by native English speaker, hopefully the manuscript is better than before.

The revised manuscript has been once resubmitted to your journal. We look forward again to your positive response. Should you have any doubts, please do not hesitate in contacting us.

Looking forward to hearing from you.

Yours sincerely

S.Chanthai

(Prof. Dr. Saksit Chanthai)

Reviewer 3 Report

The manuscript entitled: Highly sensitive fingerprint detection under UV light on non-porous surface using starch powder based luminol doped carbon dots (N-CDs) from tender coconut water as a green carbon source needs some improvements to be accepted

  1. The space between words must be revised.
  2. Here is need "at" in abstract :easy to detect under UV lamp at 395 nm.
  3. The figure after the abstract needs a caption, and the fluorescence spectra must be enlarged.
  4. In the introduction must be cited references where previously has been used NCDs such as: Carbon 144 (2019) 791-797.
  5. In my opinion the Results and discussion must be joined. The results must be discussed with the figures together.
  6. The number of figures must be decreased, and sent to a SI. Authors must be concised in this aspect, all figures cannot be showed in the main text. For instance Figure 4 and 5 are not so trascendental.
  7. Caption of figure 1 must be indicated which sample is discussed, I think is N-CDs. The XPS discussion needs to be improved, and the deconvolution peaks are not presented, must be shown. In the paper previously mentioned is well explained this technique, please use it. Is very important to do this in the case of C1s. Moreover is not inidcated the refererence used for XPS analysis (i.e 284.7 eV is used for this, the adventitious carbon).
  8. In Instrumentation is not mentioned the facilities for the characterization. It must be supplied. (Section 4.2)
  9. My main concer ins this, why is not showed the fluorescence spectra of raw luminol, when compared the carbon nanoparticles, this must be showed.
  10. The Morphology analyzed by TEM (Figure 3a) is so difficult to observe the nanoparticles, authors must show a survey image of the nanoparticles, Figure 3a must be changed.
  11. As mentioned previously, numebr of figures must decrease, not all experiments must be shown. Try to move to SI
  12. In the Application section, as mentioned previously, the experiments with luminol alone must be shown to compare the results.
  13. Why is mixed the CDs with tapioca?? and in the synthesis procedure why is mixed with C starch?? in the Graphic at the beginning of the text and not mentioned in the synthesis procedure.
  14. The Results must be validated by AFIS system (see Carbon 144 (2019) 791-797)

In my opinion this manuscript must be improved and well explained in the discussion. I do not suggest to publish it in the present form, must be improved in terms of discussion and format. Authors must detailed all the showed points, and reduce the figures, a SI anex is needed here.

Author Response

Response to reviewer’s comments

Journal: MDPI-Nanomaterial

Manuscript ID: nanomaterials-1565490

Title: Highly sensitive fingerprint detection under UV light on non-porous surface using starch powder based luminol doped carbon dots (N-CDs) from tender coconut water as a green carbon source

Dear Editor of Nanomaterial - MDPI

Thank you for your useful comments and suggestions of our manuscript. We have modified the manuscript accordingly, and the detailed corrections are listed:

Response to Comments from Reviewer 3

The authors thank the reviewer for the comments. The revision is highlighted in yellow.

Question or Comment and Answer and Suggestion:

  1. The space between words must be revised.

A1: thank you for the suggestion, hopefully the new manuscript after revised is better than before.

  1. Here is need "at" in abstract: easy to detect under UV lamp at395 nm

A2: Thank you for the suggestion, currently in the abstract part already has been changed as a suggestion

  1. The figure after the abstract needs a caption, and the fluorescence spectra must be enlarged

A3: Thank you for the suggestion, the caption for the picture after abstract has been added.

  1. In the introduction must be cited references where previously has been used NCDs such as: Carbon 144 (2019) 791-797

A4: The suggested references had been reviewed for the interpreting in the revised MS accordingly.

  1. In my opinion the Results and discussion must be joined. The results must be discussed with the figures together

A5: The author revised the manuscript by follow that suggestion,

  1. The number of figures must be decreased, and sent to a SI. Authors must be concised in this aspect, all figures cannot be shown in the main text. For instance, Figure 4 and 5 are not so transcendental.

A6: The author revised the manuscript, hopefully the revised manuscript once is better arranged than before.

  1. Caption of Figure 1 must be indicated which sample is discussed, I think is N-CDs. The XPS discussion needs to be improved, and the deconvolution peaks are not presented, must be shown. In the paper previously mentioned is well explained this technique, please use it. Is very important to do this in the case of C1s. Moreover, is not inidcated the refererence used for XPS analysis (i.e. 284.7 eV is used for this, the adventitious carbon)

A7: The author added the details of sample on the legend of Figure 1, and changed the XPS analysis picture by following deconvolution peak analysis.

  1. In Instrumentation is not mentioned the facilities for the characterization. It must be supplied. (Section 4.2)

A8: The author added the details of Characterization for Instrument,

“X-ray photoelectron spectra (KRATOS AXIS SUPRA), FTIR (Fourier Transform Infrared Spectrometer (FT-IR) Bruker TENSOR 27), EDX (Oxford Instruments X-Maxn), FIB-SEM (Helios NanoLab G3 CX) & HR-TEM (Spectrometer: Transmission Electron Microscope, Electron gun: Schottky field emission type electron gun).”

  1. My main concerns this, why is not showed the fluorescence spectra of raw luminol, when compared the carbon nanoparticles, this must be shown.

A9: Author did the measurement for new data of fluorescence of raw luminol as shown in Figure 7.

  1. The morphology analyzed by TEM (Figure 3a) is so difficult to observe the nanoparticles, authors must show a survey image of the nanoparticles, Figure 3a must be changed.

A10: The picture in Figure 3a is already the best one that obtained by analysis using HR-TEM, in this picture of the N-CDs can be see clearly the d-spacing and the diameter size of the N-CDs.

  1. As mentioned previously, number of figures must decrease, not all experiments must be shown. Try to move to SI

A11: The author revised the manuscript, hopefully the revised manuscript is better performed than before.

  1. In the Application section, as mentioned previously, the experiments with luminol alone must be shown to compare the results

A12: The reason why the author didn’t include the results from luminol alone in the application is that because pure luminol powder can’t detect the latent fingerprint clearly, and also doesn’t have fluorescence intensity under UV light as shown in the following pictures.

  1. Why is mixed the CDs with tapioca?? and in the synthesis procedure why is mixed with C starch?? in the Graphic at the beginning of the text and not mentioned in the synthesis procedure.

A13: “The author mixed the CDs with tapioca due to make a fluorescent powder to detect the latent fingerprint, the starch powder has its characteristic that can bind with the lipid on the latent fingerprint residue.” (line 188-193)

  1. The Results must be validated by AFIS system (see Carbon 144 (2019) 791-797).

A14: The author has limitation about the validation result by using AFIS system, because AFIS system in Khon Kaen, Thailand only has under Police Forensic Department 4 departments, and after the author negotiation with the Police Dept., they say that it can’t be used for any public research either.

The revised manuscript has been once resubmitted to your journal. We look forward again to your positive response. Should you have any doubts, please do not hesitate in contacting us.

Looking forward to hearing from you.

Yours sincerely,

  1. Chanthai

(Prof. Dr. Saksit Chanthai)

Round 2

Reviewer 1 Report

All the comments are not clearly seen in the manuscript. The new references should be included with proper highlight so that the reviewer could see it. Author should also mention what type error base was given in fig. 11 (is it percentage error bar or standard deviation error bar?). 

Author Response

Response to reviewer’s comments

Journal: MDPI-Nanomaterial

Manuscript ID: nanomaterials-1565490

Title: Highly sensitive fingerprint detection under UV light on non-porous surface using starch powder based luminol doped carbon dots (N-CDs) from tender coconut water as a green carbon source

 Dear Editor of Nanomaterial - MDPI

Thank you for your useful comments and suggestions of our manuscript. We have modified the manuscript accordingly, and the detailed corrections are listed:

 Response to Comments from Reviewer 1

The authors thank the reviewer for the comments. The revision is highlighted in yellow.

Question or Comment and Answer and Suggestion:

  1. All the comments are not clearly seen in the manuscript. The new references should be included with proper highlight so that the reviewer could see it. Author should also mention what type error base was given in fig. 11 (is it percentage error bar or standard deviation error bar?).

A1: thank you for the suggestion and comment hopefully the new manuscript after revised is better than before regarding the comments and new data. The error bar in fig. 11 is represented standard deviation of three independent measurements and author has been adding the detail on the revised MS accordingly.

The revised manuscript has been once resubmitted to your journal. We look forward again to your positive response. Should you have any doubts, please do not hesitate in contacting us.

Looking forward to hearing from you.

Yours sincerely,

  1. Chanthai

(Prof. Dr. Saksit Chanthai)

Reviewer 2 Report

The revised manuscript can be accepted.

Author Response

Response to reviewer’s comments

Journal: MDPI-Nanomaterial

Manuscript ID: nanomaterials-1565490

Title: Highly sensitive fingerprint detection under UV light on non-porous surface using starch powder based luminol doped carbon dots (N-CDs) from tender coconut water as a green carbon source

 Dear Reviewer 2 and Editor of Nanomaterial - MDPI

Thank you for your useful comments and suggestions of our manuscript.

Yours sincerely,

  1. Chanthai

(Prof. Dr. Saksit Chanthai)

Reviewer 3 Report

The manuscript can be accepted, after reading the responses is OK

Author Response

Response to reviewer’s comments

Journal: MDPI-Nanomaterial

Manuscript ID: nanomaterials-1565490

Title: Highly sensitive fingerprint detection under UV light on non-porous surface using starch powder based luminol doped carbon dots (N-CDs) from tender coconut water as a green carbon source

 Dear Reviewer 3 and Editor of Nanomaterial - MDPI

Thank you for your useful comments and suggestions of our manuscript.

Yours sincerely,

  1. Chanthai

(Prof. Dr. Saksit Chanthai)

Round 3

Reviewer 1 Report

The references are still not mentioned as referred by the reviewer. 

In the comment there were two updated references, '10.1021/acsami.0c14527; 10.1002/slct.202001613.'

Author Response

Response to reviewer’s comments

Journal: MDPI-Nanomaterial

Manuscript ID: nanomaterials-1565490

Title: Highly sensitive fingerprint detection under UV light on non-porous surface using starch powder based luminol doped carbon dots (N-CDs) from tender coconut water as a green carbon source

Dear Editor of Nanomaterial - MDPI

Thank you for your useful comments and suggestions of our manuscript. We have modified the manuscript accordingly, and the detailed corrections are listed:

Response to Comments from Reviewer 1

The authors thank the reviewer for the comments. The revision is highlighted in yellow.

Question or Comment and Answer and Suggestion:

  1. The references are still not mentioned as referred by the reviewer. In the comment there were two updated references, '10.1021/acsami.0c14527; 10.1002/slct.202001613.

A1: The author has been revised the new manuscript hopefully after revised is better than before regarding the comments and suggestion from the reviewer. the detali, 
- 10.1021/acsami.0c14527 in References no 33 "line 358-359"

- 10.1002/slct.202001613 in References no 52 "line 395-396"

The revised manuscript has been once resubmitted to your journal. We look forward again to your positive response. Should you have any doubts, please do not hesitate in contacting us.

Looking forward to hearing from you.

Yours sincerely,

S. Chanthai

(Prof. Dr. Saksit Chanthai)
